# The COVID-19 Mortality Rate Is Associated with Illiteracy, Age, and Air Pollution in Urban Neighborhoods: A Spatiotemporal Cross-Sectional Analysis

**DOI:** 10.3390/tropicalmed8020085

**Published:** 2023-01-26

**Authors:** Alireza Mohammadi, Elahe Pishgar, Munazza Fatima, Aynaz Lotfata, Zohreh Fanni, Robert Bergquist, Behzad Kiani

**Affiliations:** 1Department of Geography and Urban Planning, Faculty of Social Sciences, University of Mohaghegh Ardabili, Ardabil 56199-11367, Iran; 2Department of Human Geography, Faculty of Earth Sciences, Shahid Beheshti University, Tehran 19839-69411, Iran; 3Department of Geography, The Islamia University of Bahawalpur, Bahawalpur 63100, Pakistan; 4Department of Geography, University of Zurich, CH-8006 Zurich, Switzerland; 5Geography Department, Chicago State University, Chicago, IL 60628-1598, USA; 6Ingerod, SE-454 94 Brastad, Sweden; 7Centre de Recherche en Santé Publique, Université de Montréal, 7101, Avenue du Parc, Montreal, QC H3N 1X9, Canada

**Keywords:** spatiotemporal analysis, socio-economic, determinants, air quality, COVID-19 mortality, air pollution

## Abstract

There are different area-based factors affecting the COVID-19 mortality rate in urban areas. This research aims to examine COVID-19 mortality rates and their geographical association with various socioeconomic and ecological determinants in 350 of Tehran’s neighborhoods as a big city. All deaths related to COVID-19 are included from December 2019 to July 2021. Spatial techniques, such as Kulldorff’s SatScan, geographically weighted regression (GWR), and multi-scale GWR (MGWR), were used to investigate the spatially varying correlations between COVID-19 mortality rates and predictors, including air pollutant factors, socioeconomic status, built environment factors, and public transportation infrastructure. The city’s downtown and northern areas were found to be significantly clustered in terms of spatial and temporal high-risk areas for COVID-19 mortality. The MGWR regression model outperformed the OLS and GWR regression models with an adjusted R^2^ of 0.67. Furthermore, the mortality rate was found to be associated with air quality (e.g., NO_2_, PM_10_, and O_3_); as air pollution increased, so did mortality. Additionally, the aging and illiteracy rates of urban neighborhoods were positively associated with COVID-19 mortality rates. Our approach in this study could be implemented to study potential associations of area-based factors with other emerging infectious diseases worldwide.

## 1. Introduction

COVID-19 infected hundreds of millions of people and killed over 6.6 million by December 2022, and it continues to impact communities worldwide [1,2]. Since the beginning of the outbreak, Iran has had the highest number of total cases in the eastern Mediterranean region (more than 7 million, 9.4% of the total population). Additionally, the country experienced the highest total COVID-19-associated deaths (144,673; CFR 1.91%) [3,4]. For example, during the first 20 months of the outbreak of COVID-19 in Tehran, the capital and the most populated city of Iran, 60,111 infections and 7034 deaths were recorded [5].

Despite the fact that numerous COVID-19 vaccines are available, the virus continues to mutate [6,7] and spread as new variants, such as Omicron [8]. In addition, the association between COVID-19 incidence rates and biological, socioeconomic, and environmental determinants, such as air pollution, is established in the literature [9]. However, cities with large populations are economic growth centres with high levels of aggregation and mobility, making it difficult to control the rate of COVID-19 spread [6,9,10,11,12,13]. Moreover, social factors that might cause poor health increase the population’s vulnerability during pandemics [14].

Several COVID-19-related studies have focused on demographics and built environment determinates [15,16], for instance, the association between the number of patients and their residence location in terms of population and home, type of land use, and the role of underlying diseases [17]. The most effective strategy at the individual level was to limit mobility by spending more time at home and reducing visits to stores, workplaces, and public transportation [18]. Accordingly, concentrations of microscopic matter, such as PM_10_ and PM_2.5_, and various noxious air gases decreased significantly during the lockdown, resulting in improved air quality [19,20]. Demographic, socioeconomic, and climatic factors have been identified as influencing COVID-19 incidence and death rates [21,22]. The COVID-19 mortality rates have been associated with air pollution, highlighting the importance of investigating COVID-19 spread and mortality in relation to air quality [23].

Geospatial analyses contribute to the empirical study of spatial associations between the incidence of infectious diseases and socioeconomic and built environments [24] and also between pathological factors (causes, vectors, hosts, and people) and their spatial and ecological determinants [25]. Due to the limitations of the COVID-19 and risk factor data, the application of geospatial techniques was initially limited to cluster analysis via global and local Moran’s I, hotspot analysis, interpolation, and space-time scan statistics [26,27]. However, according to a recent systematic review by Nazia et al. [21], a variety of spatial analytic techniques have been used to study COVID-19 in association with various risk factors, ranging from commonly used descriptive methods (85%) to Bayesian methods (15%). While the traditional frequentist method uses the likelihood function to derive parameter estimates, the Bayesian approach incorporates probability to measure uncertainties in estimates, prediction, or inference on posterior distributions by specifying priors [21].

Compared to other methods, spatial statistical modeling of geographically weighted regression (GWR) has been widely used to identify the drivers of COVID-19 spatial variations [21]. For example, ordinary least squares (OLS) regressions ignore spatial autocorrelation and heterogeneity. Spatial regression methods, such as GWR, were used to address non-stationarity. Ganasegeran et al. [28] and Yilmaz and Ulubaş Hamurcu [29] investigated the spatial relationship between socio-demographic determinants and COVID-19 in Malaysia and Turkey using GWR. They concluded that gender, household size, the GINI coefficient [28,29], and population density are significant determinants of COVID-19 occurrence. Han et al. [30] investigated the effect of air pollution and meteorological factors on incidence rates using the generalized linear mixed effect model and GWR. They conclude that the influence of meteorological factors is greater than that of air pollution factors, and that the interaction effect of meteorological and air pollution factors on COVID-19 incidence is greater than their individual effects.

The previous studies investigated the spatial-temporal characteristics of urban neighborhoods and influential urban factors in the spread of COVID-19 in Tehran, Iran, using the GWR method [12]. They found that population density in health care facilities and public transportation hubs, such as bus stops, were associated with the spread of the COVID-19 virus. In addition, Nasiri et al. [18] found that COVID cases are more common in crowded and commercial areas. Notably, patients with co-morbidities had a higher risk of death and infection with COVID-19 than healthy people [14,18].

Given the number of available influencings area-based factors on COVID-19 mortality rates, this study aimed to look into the spatial and temporal trends of COVID-19 mortality rates in Tehran, Iran, and their associations with socioeconomic, air quality, public transportation, and built environment variables. We believe that a comprehensive examination, including more possible components, would better reveal each factor’s role, since the predictors compete. Therefore, our study has compared three different regression models, including OLS, GWR, and MGWR, to model the association of COVID-19 mortality rates with the explanatory factors. Finally, our study included 20 months of data, including four peaks of COVID-19 in Tehran, which was not covered by previous studies. Thereby, our study poses questions to obtain the objectives of this research. (1) What are the spatial, temporal, and spatiotemporal patterns of the COVID-19 mortality rates in Tehran, Iran? (2) What risk factors explain the spatially varying COVID-19 mortality rates at the neighbourhood level in Tehran?

## 2. Materials and Methods

### 2.1. Study Area

Tehran is located at latitude 35°68′92″ N and longitude 51°38′90″ E. It is divided into 22 districts and 350 neighborhoods and covers an area of 615 km^2^ [18] (Figure 1). The population is over 9 million, according to the most recent census in 2016 (4,522,000 females and 4,534,000 males). The average population density in the neighbourhoods is 21,503 people per km^2^, with a standard deviation (SD) of 12,785 [31].

Tehran’s neighbourhoods’ environmental, economic, and social characteristics are highly diverse and dissimilar [31]. Air pollution is caused by a high population density, the dominance of personal mobility, the entry of dust from the South, and the presence of polluting businesses in the West and Southwest (27 km^2^ (3.7%) of the city’s total area is devoted to industrial purposes) [12,32,33,34]. Tehran is one of the world’s most polluted cities [33], with levels of O_3_, NO_2_, and particular matter of ≤2.5-micron size (PM_2.5_) remaining high throughout the year but reaching the highest levels in the fall and winter [34,35]. Tehran’s annual average air temperature is about 15–18 °C [32].

Respiratory disorders have been a leading cause of death in Tehran over the last decade (2008–2018), accounting for 14% of all deaths [12]. Furthermore, since the outbreak of COVID-19, Tehran has had the highest COVID-19 mortality rate (20% of all daily deaths) of any Iranian city [31,36].

### 2.2. Data

COVID-19 mortality data were obtained from the Iranian Ministry of Health and Medical Education’s Hospital Information System (HIS) [36]. We excluded data that were incomplete or inaccurate (about 50 records). Our analysis was based on data from 7,043 people who died due to COVID-19. This dataset spans a period of twenty months, covering December 2019 to July 2021 and includes the date of death, age, gender, hospitalization location, and date and home address [36,37].

This study’s administrative geographical data include the most recent municipal division data (city, region, and neighbourhood boundaries). The built environment and land statistics were obtained from the Tehran city municipality [36]. Spatial analytic methods were used to produce additional spatial data, such as spatial density indicators for our study variables (Appendix A). The study variables were aggregated at the neighbourhood level using ArcGIS Pro 3.0.2 software (ESRI, Redlands, CA, USA).

The Tehran Air Quality Control Company web portal was used to obtain air quality index (AQI) data from 2016 to 2021 [32]. The public transport data were obtained from the municipality of Tehran directly and updated with Open Street Map (OSM) datasets (https://www.openstreetmap.org/ accessed on 23 January 2023) using the QGIS platform (https://www.qgis.org/ accessed on 23 January 2023). In addition, the Iranian Statistical Centre provided the socioeconomic characteristics of neighborhoods (including unemployment rates) of 2016 [38].

### 2.3. Explanatory Variables

The explanatory variables used in this study are listed in Table 1. Figure 2 shows how these variables are distributed spatially across the study area.

(1)Built environment, land use, and urban facilities: previous research has found a relation between the density of various land uses and the COVID-19 epidemic [38,39]. The goal of this study, on the other hand, was to determine the spatial association between land use categories and COVID-19 mortality rates. As a result, the density of various uses, such as banks, restaurants, and high-rise residences, was investigated.(2)AQI: the literature has shown a strong, positive relationship between air pollutants and COVID-19 transmission and mortality in several geographic regions [40,41,42]. The major air contaminants considered as initial independent variables in this study are NO, NO_2_, O_3_, CO, and particulate matter PM_2.5_ and PM_10_ (NO and NO_2_ are collectively referred to as NO_x_). Some previous studies have used the effects of these pollutants on respiratory diseases individually or in combination with each other [23,43,44,45,46,47,48,49,50]. In the present study, both were used, and the most significant variables were included in the final model. Exposure to air pollution is an essential risk factor for many of the chronic diseases that cause people to be more likely to become seriously ill, require intensive care and mechanical ventilation, and die from COVID-19 [51]. In this study, the average of 5 years (2016 to 2021) for each pollutant was derived from the pollutant-related data of air pollution monitoring stations. Then, using the inverse distance weighted (IDW) interpolation technique, data were calculated in a GIS with pixels of 1 × 1 km in size. The 5-year average of pollution was then calculated for each neighbourhood separately using zonal statistical methods.(3)Public Transportation: public transportation contributes to the geographic spread of COVID-19 [47]. Few studies have examined the correlation between these factors and mortality rates. To analyze the association at the neighbourhood level, we considered variables such as distance from fuel stations, metro and bus rapid transit (BRT) stations, and the spatial density of main roadways.(4)Socioeconomic features: recent research has shown that socioeconomic variables impact COVID-19 transmission and mortality in various settings in developing countries, such as Iran [52]. In this study, we looked at the spatial correlations between six such variables (population density, illiteracy, unemployment, older age, having a rented home, and being an immigrant) and neighbourhood-level COVID-19 mortality.

**Table 1 tropicalmed-08-00085-t001:** Potential explanatory variables and data sources.

Theme	Variable	Measure and Unit	Descriptions and Rationale	Data Source *
**Built-environment, land use, and facilities**	Commercial land use (V_1_)	Spatial density of commercial properties per km^2^	Locations with high connectivity, high density, and geographic concentration of economic activity may be at a relatively higher risk of COVID-19 infection [53].	1
	Industrial land use (V_2_)	Spatial density of industrial units per km^2^	Large numbers of industrial units in a region has a significant effect on the number of active COVID-19 cases in that region [53,54].	
	Land use for social services (V_3_)	Spatial density of social services units per km^2^	Social service centres may be a place of the spread of infectious diseases as they attract many people at times of epidemics [12].	
	Banking (V_4_)	Spatial density of banks per km^2^	Some studies have shown that banks and automated teller machines (ATMs) are important for the spread of COVID-19 [55].	
	Health service (V_5_)	Spatial density of health service centres (including special hospitals for COVID-19 patients, public clinics, and laboratories) per km^2^	For example, hospitals for COVID-19 patients, public clinics, and laboratories naturally have an extremely strong association with the rate of COVID-19 infection [12].	
	Deteriorated buildings (V_6_)	The ratio of areas with deteriorated and old buildings to the total area of each neighbourhood in km^2^ * 100 (%)	Low-income and impoverished people generally reside in worn and inadequately constructed settings, where the houses are deteriorated leading to a relatively great danger of disease outbreak [12,56].	
	High-rise buildings (V_7_)	Spatial density of residential high-rise buildings per km^2^	Overcrowding, dense space, and health conditions in high-rise buildings can increase the risk of COVID-19 outbreaks, which can affect people in different age groups and individuals who are suffering from underlying diseases [57].	
	Distance from the city business district (CBD) (V_8_)	Distance to the CBD in km	Previous studies show that the COVID-19 transmission decreases with the distance from the city centre [47].	
	Presence of restaurants (V_9_)	Spatial density of restaurants per km^2^	Controlling transmission in restaurants is an important component of public health measures for COVID-19 [58].	
**Air Quality Index**	Particulate matter of size ≤2.5 micron (PM_2.5_) (V_10_)	Spatial density of particles ≤2.5 micron (PM_2.5_) per m^3^ of air (µg/m^3^)	Studies have shown a positive correlation between the effect of delayed PM_2.5_ concentration and the number of confirmed COVID-19 cases, indicating an increased risk of infectious diseases [59,60,61].	2
	Particulate matter of size ≤10 micron (PM_10_) (V_11_)	Spatial density of particles ≤10 micron (PM_10_) per m^3^ of air (µg/m^3^)	Studies have confirmed that new cases of COVID-19 are associated with elevated PM_10_ concentrations in urban areas [45,46].	
	Carbon monoxide (CO) (V_12_)	Concentration of carbon monoxide (CO) in parts per million (ppm)	Most studies confirm that both COVID-19 cases and deaths are positively associated with almost all pollutants [44].	
	Nitrogen dioxide (NO_2_) (V_13_)	Concentration of nitrogen-dioxide (NO_2_) in parts per billion (ppb)	Studies have shown that there is a significant association between NO_2_ and the risk of COVID-19 infection [23,43].	
	Nitrogen monoxide (NO) (V_14_)	Concentration of nitrogen monoxide (NO) in parts per billion (ppb)	Studies have mentioned the role of the NO pollutant in COVID-19 transmission and death [44,45,46,47].	
	(V_15_) Nitrogen oxides (NO_x_ ppb)	Concentration of nitrogen oxides ((NO+NO_2_) parts per billion (ppb)	Some studies have demonstrated that NOx (NO+NO_2_) emissions significantly increase the incidence of COVID-19 transmission and death [48,49,50].	
	Ozone (O_3_) (V_16_)	Concentration of ozone (O_3_) in parts per billion (ppb)	Studies have demonstrated that COVID-19 outbreaks and fatalities are associated with ozone levels [44,45,46,47].	
	Sulfur-oxide (SO_2_) (V_17_)	Concentration of sulfur oxide (SO_2_) in parts per billion (ppb)	This gas is released by airplanes, trains, and other means of transportation. The importance of reducing it during quarantine situations have been highlighted [62].	
	Temperature (V_18_)	Average annual temperature (2011-2021) in degrees Celsius (°C)	High temperatures increase the risk of COVID-19 diseases and are associated with death from respiratory diseases, as well COVID-19 [47,63,64].	
**Public Transportation**	Metro stations (V_19_)	Distance to the metro stations in meters	Many studies highlight the role of public transportation in the spread of infectious diseases, even in remote areas. Research shows that public transportation facilities play an important role in the geographical spread of COVID-19 [23,55,65].	3
	Bus rapid transit (BRT) stations (V_20_)	Distance to the BRT stations in meters	Sense places, such as BRT stations, are at risk for many physical contacts and disease transmission [66,67].	
	Density of roads (V_21_)	Spatial density of main roads per km^2^	Where the urban road network creates the most intersections, individual and collective contacts increase. In the long run, this increases the spread of the disease in the surrounding areas [68].	
	Fuel stations (V_22_)	Distance to the fuel (petrol and gas) stations in meters	As with other location based public facilities, fuel stops may increase the transmission and spread of the COVID-19 virus in nearby areas [55].	
**Socio-economic characteristics**	Population density (V_23_)	Total population/neighborhood area (km^2^) = persons/km^2^.	There is a significant relationship between population density, overcrowding, and the spread of COVID-19 virus [64,69].	4
	Illiteracy rate in % (V_24_)	Ratio of illiteracy in the total population ≥ 6 years = illiterate population/population (6+) * 100, (%).	Health literacy enables people to understand the reasons behind medical recommendations and to become aware of the possible outcomes of their actions. Instead, higher levels of adult’s illiteracy rates can be seen as a social risk factor for rising COVID-19 related deaths [70].	
	Unemployment rate in % (V_25_)	Ratio of unemployment in the total population = unemployed population/active population (15–65 years) * 100 (%).	Areas with a higher unemployment rate positively associated with COVID-19 high mortality rates [71].	
	Age rate in % (V_26_)	Ratio of elderly in the total population = elderly (65+ years) population/total population * 100 (%).	Older age groups experienced higher COVID-19 mortality rates. Subsequently, areas with a large proportion of elderly people face a high risk of infection [72,73].	
	Rate of rented homes in % (V_27_)	Total number of rented houses/total number of all types of housing units * 100 (%)	Mostly recent studies have examined the correlation between COVID-19 outbreaks and poor housing condition [74,75].	
	Rate of Immigrants in % (V_28_)	Ratio of immigrants in the total population = immigrant population/total population * 100 (%).	Areas with higher rates of immigration appear to have been more affected by COVID-19 [76,77].	

* Sources 1: Municipality of Tehran [78,79]; 2: Tehran Air Quality Control Company (AQCC) [32,33]; 3: Open Street Map [80]; 4: Statistic Center of Iran [38].

### 2.4. Methods, Tools, and Procedure

#### 2.4.1. Geographical Smoothing of Mortality Rates

Intrinsic variance instability in estimating death due to population variation in spatial units has attracted widespread attention in disease mapping [81,82]. The spatial variance of the COVID-19 mortality rate per spatial unit requires spatial smoothing. To address this issue, we used empirical Bayesian smoothed (EBS) [82] technique in GeoDa software (Center for Geospatial Analysis and Computation, Tempe, AZ, USA) to create smoothed EB rates (per 100,000 people) to reduce random fluctuations due to population size by calculating risk as the total weighted crude rate for each neighbourhood [69,83].

#### 2.4.2. Spatio-Temporal Analysis

Before conducting a local spatial cluster analysis of COVID-19 deaths, Global Moran’s I [66] was used to investigate the global spatial distribution pattern of COVID-19 mortality rates in the study area. Kulldorff’s scan statistics approach [84] was then used to identify significant temporal, spatial, and spatio-temporal clusters of COVID-19 fatalities. The relative risk (RR) and the log-likelihood ratio (LLR) were computed. Monte Carlo simulations, first introduced by Dwass, [83] were utilized to calculate a *p*-value. Monte Carlo is one of the constituent methods for interpreting scan statistics results [85]. The maximum window size for spatiotemporal analysis was set at 50% for the study area and time using SatScan software (M Kulldorff and Information Management Services Inc, Cambridge, MA, USA). A circular shape was chosen to scan and detect all significant spatio-temporal clusters [86]. A Poisson model was used for COVID-19 mortality calculations [85]. For spatio-temporal and purely temporal models, the time aggregation period was set at one month [83]. No geographical overlap was defined as the criteria for reporting secondary clusters. The Appendix A explain the purely temporal, purely spatial, and spatiotemporal results of the scan statistics methodology.

The Monte Carlo testing was set to calculate test statistics for each random replication at the *p* = 0.05 level. Irrespective of the number of Monte Carlo replications chosen, the hypothesis is unbiased, resulting in a correct significance level that is neither conservative nor liberal or an estimate. In the standard Monte Carlo method with 999 random replicas, the lowest *p*-value the testing can report is 1/(999 + 1) = 0.001, which was set for calculating *p*-values in all Spatio-temporal analyses. In this study, a Monte Carlo test with 1,000 iterations was applied to evaluate the spatial variability of each surface of parameter estimates produced by the multiscale geographically weighted regression (MGWR) model [87]. All the spatial results were mapped using QGIS, V.3.26, a free and open-source GIS package [88].

#### 2.4.3. Linear and Geographically Multivariate Data Analysis

Pearson’s correlation coefficient and R^2^ correlation were used to explore the global and linear correlation between COVID-19 mortality rates (per 100,000 population) and exploratory variables. Mirrored scatter plot matrix was used to visualize the bivariate relationships between combinations of variables. Each scatter plot in the matrix visualizes the relationship between a pair of variables [89]. According to Pearson’s coefficient and R^2^ values, five variables (including V7, V23, V25, V27, and V28) were removed from the list of explanatory variables (Appendix A for details). Mirrored scatter plot matrix shows the global relationships between COVID-19 mortality rates and all selected explanatory variables in Tehran based on Pearson’s correlation test (Appendix A). The methodology flowchart of this research is shown in Figure 3.

In the following step, exploratory regression was used to model linear relationships and select the most important variables for the OLS analysis while considering multicollinearity. While exploratory regression is similar to stepwise regression, rather than only looking for models with high adjusted R^2^ values, exploratory regression looks for models that meet all of the requirements and assumptions of the OLS model. The variance inflation factor (VIF) indicates multicollinearity, and values between 5 and 10 are recommended, with 10 as the maximum VIF level [86,90]. The max VIF was set at 7.5, as ESRI recommended [91,92], and the model was executed eight times to select the final significant variables. Based on these exploratory regression results, the variables V11 (PM_10_), V13 (NO_2_), V16 (O_3_), V24 (illiteracy rate), and V26 (proportion of elderly) as variables with the highest significance (≥60%), which explain the highest adjusted R^2^ (≥0.5), were included in our OLS model (Appendix A). Based on these values, an OLS multivariate regression model was examined to explore the spatial autocorrelation of residuals (see Appendix A for details). The technique assumes a stationary and constant relationship over space [92]. Using Moran’s I, we looked for the presence of OLS residuals and spatial autocorrelation within the study area. The value of Moran’s I varies between +1 and −1 [64], where a value close to 1 represents strong spatial autocorrelation [93]. We then employed the GWR model (Appendix A for details), an extension of the basic OLS standard regression, reflecting the variables’ distribution’s spatial heterogeneity [94]. The model can estimate and generate a set of local parameters, including adjusted R^2^, the corrected Akaike’s information criterion (AIC_c_), local coefficients, and residuals for each spatial unit to examine the spatial variation of the relationship between response and explanatory variables [95].

In the GWR model, the regression parameters of a particular location carry out local regression estimates based on sub-sample information for adjacent areas. The parameters of the estimated variables are adjusted as the spatial position changes [95]. GWR provides advantages to regression modelling and accounts for spatial variations, considering the spatial scale constant over time and space. However, in many cases, a fixed spatial scale is not valid where phenomena involve numerous spatial processes with various spatial scales [96]. Accordingly, MGWR is used to explore the relation between the response variable and exploratory variables, whichever vary spatially at different scales (see Appendix A for details). Compared to GWR, the MGWR model has several advantages. Particularly, it can accurately depict spatial heterogeneity, diminish collinearity, and lessen the bias in the parameter estimates [97]. Each predictor variable has its bandwidth in an MGWR model, which allows the scale of non-stationary relation to vary for each response to the predictor variable relation, as described in equation (Equation (1)), assuming that there are n observations, for observation i∈{1,2,…,n} at location $\left(u_{i},v_{i}\right)$,
(1)$$y_{i}=\beta_{0}\left(u_{i},v_{i}\right)+\overset{m}{\underset{j=1}{\sum }}\beta_{bwj}(u_{i},v_{i})x_{ij}+\varepsilon_{i}$$
where $\beta_{0}\left(u_{i},v_{i}\right)$ is the intercept; $x_{ij}$ is the *j*th predictor (independent) variable in the coordinate of each observation $\left(u_{i},v_{i}\right)$; $\beta_{j}(u_{i},v_{i})$ is the *j*th coefficient; $\varepsilon_{i}$ is the error term; and *y*^i^ is the response variable, while *bwj* in $\beta_{bwj}$ indicates the bandwidth used for the calibration of the *j*th conditional relationship [98]. The open-source MGWR application (Python) was used to run the GWR and MGWR models [99]. The Gaussian model was used to run both models and locations (identified by identification numbers), coordinates variables (x and y), four independent variables (Appendix A), and the EBS mortality rate as the dependent variable were introduced to the models. To select optimal bandwidths in both models for comparison purposes, the adaptive bi-square spatial kernel method [100] was used, and the Golden Section mode [100] was applied as a weighting scheme for calibrating both models. The AIC_c_ was used as an optimization criterion in the calibration of the GWR and MGWR models. In addition, local VIFs [100] were applied to evaluate probable multicollinearity among explanatory variables at different spatial units. It was also possible to test the statistical significance of each surface of parameter estimates produced by GWR and MGWR via random sampling. In this study, a Monte Carlo test with 1000 iterations [101] was applied to evaluate the spatial variability of each surface of parameter estimates produced by the MGWR model. A pseudo-*p*-value <0.05 indicates that the observed spatial variability of a coefficient surface is significant at 95% CI (i.e., non-random) [100].

This study used *Pseudo-t* statistics and explanatory variable standardized coefficients (*Beta*) to map the spatially varying relationships between COVID-19 mortality rates and the explanatory variables. Bivariate choropleth maps, which show the quantitative relation between two variables in a feature layer [100], were used to represent and compare the MGWR model parameter estimates and original values of the explanatory variables (Fig.10). This mapping method is useful for finding the local patterns and variations of two parameters in a single map [92]. The ArcGIS Pro 3.0.2 package (ESRI, Redlands, CA, USA, 2022) was used to visualize our final model results.

## 3. Results

### 3.1. COVID-19 Mortality Rates

Figure 1 shows the spatial distribution of Tehran’s COVID-19 EBS mortality rates (per 100,000 population) in Tehran. Among the 350 neighbourhoods investigated, the lowest EBS rate was about 11, and the highest was about 350 (mean = 85 and SD = 50). This map shows that most high-rate neighbourhoods are located in the central and southern parts of the study area.

During the 20-month study period, 60,111 individuals reportedly contracted COVID-19 in the study region. In addition, data analysis reveals that COVID-19 caused the deaths of 7043 people (an average of 78 per 100,000 population). The monthly distribution of the mortality rate per 100,000 population by gender is depicted in Figure 4a. This graph depicts the variation in mortality rates between the two sexes over twenty months. In certain months (e.g., February 2019, September and October 2020 and March 2021), the monthly mortality rate for both groups reached as high as 8 per 100,000 inhabitants. Importantly, male mortality rates (4.57 per 100,000 men on average) were higher than female rates (3.27 per 100,000 women on average). Figure 4b displays the COVID-19 mortality rate by gender and age group. Men still account for 58.9% (91 deaths per 100,000 men) of the total COVID-19-related deaths, whereas women account for 41.1% (64 deaths per 100,000 women). However, the mortality rate was 68% higher among those aged ≥65 years, while those under 25 years had the lowest COVID-19 mortality rates in Tehran, with less than one death per 100,000 population.

### 3.2. Spatio-Temporal Distribution Patterns and Clusters

#### 3.2.1. Retrospective Purely Temporal Analysis

The purely temporal pattern of COVID-19 deaths is shown in Figure 5. Four peaks of COVID-19 deaths over the study period were observed. COVID-19 deaths peaked in September and October 2020 and from March to April 2021. In addition, statistically significant clusters (LLR value = 523.4, *p* < 0.05) were formed during the months of February 2020 to November 2020.

#### 3.2.2. Global and Local Purely Spatial Analysis

The results of Moran’s I (I = 0.143 and *p* < 0.05, z-score = 4) show that the COVID-19 mortality rates (from 2019 to 2021) were spatially clustered in the study area. This result allowed us to enter local-scale spatial analyses more confidently. Based on purely spatial statistical analysis, seven significant spatial clusters (*p* < 0.05) were identified, which are given in Table 2 and Figure 6. According to these results, with a highest LLR value of 242.88 and a RR value of 1.85, cluster number one has been identified as the most likely in the study area. This cluster formed in the centre of the study area and consisted of 71 neighbourhoods with a population just above 1.5 million (number of cases = 1956 and number of expected cases = 1210). As illustrated in Table 2 and Figure 6, cluster number 3 (LLR = 26.2 and RR = 1.47) is another important cluster formed in the northern part of the city covering 14 neighbourhoods and 421 deaths.

#### 3.2.3. Retrospective Space-Time Analysis

As Figure 7 shows in more detail, we identified only one significant space-time cluster (LLR = 467.31 and RR = 2.29, *p* < 0.05). This cluster consists of 75 neighbourhoods with a population of about 3,567,000. In addition, this cluster formed from February to October 2020 (nine months), which corresponds to the temporal clustering pattern (Figure 5). This cluster extends from the centre to the south and southeast areas of the city.

### 3.3. Modeling Spatial Associations

#### 3.3.1. OLS Model Results

The results of the OLS model show that, with an AIC_c_ value of 3,455.82, and R^2^ value of 0.57 and an adjusted R^2^ value of 0.56, our model accounts for about 57 and 56% of the COVID-19 mortality rates in the study area. Furthermore, the results demonstrate that the probability and robust probability coefficients of our independent variables (selected by exploratory regression) were also statistically significant (*p* < 0.01) in the OLS model (Appendix A). When Moran’s I was applied to ensure that the OLS model residual values really were spatially random, the result showed that the autocorrelation was significant, and residuals were spatially clustered (Moran’s I = 0.160. z-score = 6.5, *p* < 0.05). However, as follows, GWR and MGWR can solve the problems of non-stationarity and problems with heteroscedasticity. In addition, these models enabled us to improve the reliability of the predictions [92].

#### 3.3.2. GWR Model Results

Diagnostic indicators of the GWR model showed that, with the AIC_c_ value of about 661, the R^2^ value of 0.67, and the adjusted R^2^ value of 0.65, our GWR model accounts for about 67 and 65% of the COVID-19 mortality rates within the study area (Table 3). The statistics for the GWR model are summarized in Table 4, which demonstrates that the GWR model predicts changes in local coefficients at the local scale but within the same bandwidth. The MGWR model can address this weakness of the GWR model by allowing various bandwidths (Table 4). The column “Mean” in Table 4 shows the general coefficients of both GWR and MGWR models. For instance, NO_2_ had the strongest association with COVID-19 mortality rates in the whole city, regardless of any specific geographical area.

#### 3.3.3. MGWR Model Results

The diagnostic metrics of the MGWR model are provided in Table 5. The results of the MGWR model showed that, with the AIC_c_ values of about 661 and the R^2^ value of 0.69, an Adj R^2^ value of 0.66 in our model accounts for about 69 and 66% of the COVID-19 mortality rates within the study area (Table 5). In addition, as indicated in Table 4, the MGWR model improved the predictions of the estimate of the explanatory variable’s coefficient in the local linear regression model by varying the bandwidth of each explanatory variable [92].

#### 3.3.4. Model Performance (Validation)

The summary of the model’s diagnostics shows that the adjusted R^2^ value increased from 0.56 (OLS) and 0.64 (GWR) to 0.66 (MGWR). Moreover, selecting the most parsimonious model, AIC_c_ and the residual sum of squares (RSS) with the lowest values are preferred [92]. As Table 6 shows, AIC_c_ decreased from 3455.81 (OLS) and 660.788 (GWR) to 641.051 (MGWR). The MGWR model produced a better AIC_c_ (641.051), which indicates that the MGWR model provides significantly better goodness-of-fit than OLS and GWR models when assessing the relationship between the explanatory variable and the COVID-19 mortality rates.

Furthermore, to compare the difference in the goodness of fit of the GWR and MGWR models for different spatial units, the local R^2^ values were mapped for the GWR and MGWR models (Figure 8). These maps indicate that the GWR and MGWR models accurately explain the local relationship between COVID-19 mortality rates and dependent variables all over the city (local R^2^ > 0.6). Two models were best fitted to the city’s western, south-western, and north-western parts (Figure 8A,B). However, as shown in Figure 8C,D, the highest local R^2^ values in the MGWR model (mean of R^2^ = 0.68) covered more neighbourhoods in the city. Therefore, this study identified the MGWR model as the best model to visualize and interpret the spatial associations between COVID-19 mortality rates and dependent variables.

To examine whether the residuals of the MGWR model as the best-fitted model were autocorrelated, Anselin Moran’s I statistic was used. The results showed no significant autocorrelation in the MGWR residuals (I = −0.031, z-score = −1.12, *p* > 0.05), which is a random pattern confirming their independence. While the difference between the results of the GWR and MGWR models was not remarkable, it is reasonable that we represent the results of the best model. After selecting the best-fitted model, the estimated coefficients for each explanatory variable were mapped to display their effects on COVID-19 mortality rates across Tehran.

### 3.4. Mapping and Spatial Analysis of the MGWR Model

#### 3.4.1. Local t-Values

Figure 9 shows the spatial distribution map of the pseudo t-values of significant positive local estimates based on the MGWR model for five significant covariates. The PM_10_ variable (min = 2.6, mean = 2.85, max = 3.1, and *p* < 0.05) explains the most significant correlation with mortality rates in the western parts of the city (Figure 9-V11). The NO_2_ variable (min = 4.8, mean = 5.9, max = 6.5, and *p <* 0.05) explains the significant correlation with COVID-19 mortality rates in the western half of the city (Figure 9-V13). The relationship between O_3_ and COVID-19 mortality rates was significant (min = 3.9, mean = 5.6, max = 6.9, and *p* < 0.05) in the central neighbourhoods, which stretch toward the southern parts of the city (Figure 9-V16). As the map shows, the most associated illiteracy rate (%) (min = 0.226, mean = 2, max = 3.55, and *p <* 0.05) was observed in the south-eastern to the central and northern parts of the city (Figure 9-V24). The relationship between the age rate and COVID-19 mortality rate was (min = 6.2, mean = 6.5, max = 6.7, and *p* < 0.05) positively significant with high values in and around the central parts of the city (Figure 9-V26).

#### 3.4.2. Local Parameter Estimates

Figure 10 shows the spatial distribution bivariate maps of the local spatially varying *Beta* coefficients (standardized parameter estimates) and the values of the explanatory variables aggregated at the neighbourhood scale. These bivariate maps show that the correlation between each variable and the dependent variable was not constant across the city.

Figure 10-V11 shows the estimated coefficients for the PM_10_ (min = 0.103, mean = 0.115, max = 0.13, and *p* < 0.05). According to this map, the *Beta* values are high in 43.4% of neighbourhoods located in the north-eastern and eastern parts of the city, which indicates a strong association between PM_10_ and COVID-19 mortality rates. Although the PM_10_ values were higher in the city’s northern parts, the coefficients are low in these areas.

Figure 10-V13 shows the coefficients for the NO_2_ variable (min = 0.31, mean = 0.37, max = 0.41, and *p* < 0.05). *Beta* values of NO_2_ were higher than the mean in 88.6% of the neighbourhoods, with most of them in the eastern parts of the city. In the neighbourhoods located in the south-eastern and central neighbourhoods, the higher values of NO_2_ and *Beta* increased in parallel. However, the northern and eastern neighbourhoods of the city represent different spatial patterns. While NO_2_ values were higher in these areas, the corresponding *Beta* values were lower, which indicate weak correlations.

Figure 10-V16 shows the coefficients for the O_3_ variable (min = 0.236, mean = 0.33, max = 0.42, and *p* < 0.05). *Beta* values of O_3_ are higher than the mean in 28.2% of the neighborhoods, most of which are in the central parts of the city. Central neighborhoods had the higher values of O_3_ and *Beta* increased in parallel. However, the northern and north-western neighbourhoods of the city represent different spatial patterns. While O_3_ values were higher in these areas, the corresponding *Beta* values were lower, which indicates weak correlations in these neighbourhoods.

Figure 10-V24, shows the coefficients for the illiteracy rate (%) variable (min = 0.013, mean = 0.125, max = 0.24, and *p* < 0.05). *Beta* values of the illiteracy rates (%) were higher than the mean in 51.7% of the neighbourhoods, with most of them in the central parts toward the northern parts of the city. In the neighbourhoods located in the south-eastern neighbourhoods, the higher percentages of illiteracy rates and *Beta* increased in parallel. However, eastern neighbourhoods of the city represent different spatial patterns, too. While illiteracy rate values are higher in these areas, *Beta* values for the illiteracy rates (%) were lower, which shows the weak correlations in these neighbourhoods.

Figure 10-V26 shows the coefficients for the aging rate (%) variable (min = 0.365, mean = 0.374, max = 0.38, and *p* < 0.05). *Beta* values of the age rate were higher than the mean in 51.4% of the neighbourhoods in the central part of the city. In the neighbourhoods located in the central part of the city, the higher age groups and *Beta* increased in parallel. However, the northern and north-eastern neighbourhoods of the city represent different patterns. While ages are higher in these areas, *Beta* values for this variable were lower, which shows the weak correlations in these neighbourhoods. In fact, age had a strong correlation with COVID-19 mortality rates in the central neighbourhoods.

## 4. Discussion

We identified one purely temporal cluster of COVID-19 mortality rates in the research area between February and November of 2020 (Figure 5). Similar temporal groupings during that era have been observed in previous studies, for example, in Brazil [102]. Iran has previously dealt with a variety of diseases, but COVID-19 stunned the system with its severity, rapid spread, and pathological consequences [103]. COVID-19 vaccination did not begin until late February 2021. As a result, many people died during the following three disease peaks [104]. We discovered seven significant spatial mortality clusters, the most significant of which were Cluster 1 in the city’s center and Cluster 3 in the north. In addition, there was a significant central space-time cluster that expanded in the south and southeast (Figure 7). This cluster was formed from February 2020 to October 2020, which is extremely comparable to the COVID-19 purely temporal cluster in the research region (Figure 5). As a result of this variability and clustering, COVID-19 mortality clusters might have been influenced by the particular characteristics of the various metropolitan districts that were our second aim in this study.

Previous research indicates that environmental problems, such as air pollution concentration in Tehran, can increase COVID-19 mortality rates in big cities [36]. This is in accordance with our findings. However, the MGWR model (the best-fitting model) showed that the COVID-19 mortality rates associated with explanatory variables varied substantially across the study area. Most of the factories (e.g., industrial sand and cement factories) are in the west. Therefore, PM_10_ levels are higher in the west. This higher PM_10_ levels are also strongly associated with COVID-19 mortality in the west. However, there are some areas in the northern part of the city in which the amount of PM_10_ is not very high but is highly associated with COVID-10 mortality. Lowering the PM_10_ in these areas might also help reduce the COVID-19 mortality rate. Although the lockdowns reduced the PM_10_ levels in Tehran between 20 and 30 percent [105], this short-term drop did not diminish the citywide long-term detrimental consequences of PM_10_. Indeed, the industrial plants surrounding the city increased their production of detergents and hygiene products during the COVID-19 pandemic, which account for the continued city pollution [106]. In addition, the predominant wind direction in Tehran is from the west and south, which may carry PM_10_ and other pollutants from their sources (e.g., industrial sources, construction sites, landfills, and desert areas in the south of the city) [107]. In addition, as shown in Figure 1, South Tehran borders the country’s central deserts from where the wind carries dust and air pollutants. At the same time, the northern part of the city is surrounded by mountains, which prevent any further discharge of pollution.

In our investigation, NO_2_ was most strongly related to COVID-19 mortality rates in the eastern half of the city (Figure 9), where the concentration of this gas also is greater than in the rest of the city (Figure 10). Previous research has validated the association between NO_2_ and COVID-19 mortality rates [108,109]. Over 75% of Tehran’s residents use non-standard fuel-powered automobiles, which account for 40% of the city’s air pollution, including NO_2_ [105,109,110]. Except for the northern portion of the city, bus terminals are another source of NO_2_ across the city [106]. The long-term exposure of Tehran’s residents to air pollution has resulted in a considerable increase in the mortality rate from respiratory disorders, according to earlier research [105] and we can now see how this has exacerbated the COVID-19 situation. Several research results indicate that during the COVID-19 pandemic, public transit usage did not drop. Instead, the use of private cars grew leading to a rise in pollution levels [63,107]. For example, the Tehran Air Quality Control Company (AQCC) confirmed that from March 2021 to November 2021, Tehran was one of the most polluted cities in Iran [32].

In accordance with earlier studies that revealed the association between ozone and high COVID-19 mortality rates [33,45,108], our study demonstrated that this gas is positively associated with COVID-19 mortality rates, which we particularly noted in the central and southern parts of the city (Figure 9). In these parts of Tehran, the annual temperature is high (Figure 2), and private vehicles are commonly used due to inadequate public transportation and traffic congestion [111], which contributes to the levels of ozone, a secondary pollutant associated with high levels of other climatic factors, such as temperature and nitrogen oxides [50,112]. In addition, another study conducted in Tehran revealed that the O_3_ concentration did not decrease during the lockdowns, and this gas is recognized as one of the risk factors that raises the likelihood of COVID-19 mortality [12].

In addition to the air quality indices, our data revealed a varied positive association between illiteracy and COVID-19 mortality rates. Previous research has proven the positive correlation between socioeconomic statuses, such as illiteracy, and COVID-19 mortality [105]. While we found a correlation between COVID-19 mortality and illiteracy, which was exceptionally high in the south-western and the central regions of the city (Figure 2), there was an even higher correlation in the centre and the south-eastern neighbourhoods extending into additional northern locations (Figure 10). Two studies conducted in the United States found that a lower level of literacy led to less awareness about the risk of COVID-19 [113,114]. As a result, preventive measures are less prevalent in those populations [113]. Other research has demonstrated that COVID-19-related mortality rates are twice as high in disadvantaged areas of major cities in developing countries [114].

Men were more likely to die from COVID-19 than women [115,116,117,118]. This general gender distribution was also seen in Tehran, with about a 60 to 40% difference between males and women, respectively, throughout the twenty-month research period. According to most research, men’s COVID-19 mortality rates are affected mainly by smoking and alcohol, which affect the lungs. In addition, recent studies have shown that women have a better immune system against infections than men [119]. In addition, numerous studies demonstrate that oestrogen is a protective factor against numerous viral infections [120,121], and this hormone has been shown to suppress SARS-CoV replication by modulating cell metabolism [113]. Our findings also reveal that approximately 68% of the total COVID-19 deaths occurred in the elderly (Figure 4). Numerous prior studies indicate that older age is one of the risk factors for increased COVID-19 mortality rates [116,122,123,124]. Our findings also indicate that older age is substantially linked to COVID-19 mortality rates (Figure 9). This was particularly seen in the centre of the city where people are generally of an older age (Figure 10). Our survey reveals that 8% of the city’s population is currently ≥65 years, with the majority residing in the city’s northern regions (Figure 2 and Figure 10). Many older adults cannot take care of their health properly and are also subject to the harmful air quality in central Tehran. The elderly also suffer from a higher rate of chronic diseases (diabetes, blood pressure, lung problems, etc.), which are associated with a higher risk of death due to COVID-19 [118].

### 4.1. Policy Implications

The following long-term urban health policy proposals can help to reduce mortality rates associated with pandemics such as COVID-19.

(1) While achieving sustainable urban growth without air pollutants may be difficult, the level of urban pollution must be reduced to prevent additional respiratory illnesses and COVID-19-related deaths. A network of environmentally friendly transportation should be developed. The number of private automobiles that use fossil fuels might be reduced and replaced with electric vehicles.

(2) The old polluting industries that surround Tehran must be upgraded with cutting-edge technology and, if possible, partially replaced by "green industry." The green industry is an emerging concept that aligns industrial development with global sustainable development toward a green economy [125].

(3) It is critical to prevent the spread of respiratory disorders induced by PM_2.5_ and PM_10_ pollution in southern and western Tehran. Additionally, certain health interventions should be implemented in areas with a high probability of PM_10_ concentration. The use of facemasks, for example, is an excellent short-term solution.

(4) The elderly are more likely to suffer from COVID-19 symptoms. Therefore, short- and long-term health strategies for protecting the elderly and providing health care in high-risk areas deserve special consideration. In addition, a better urban infrastructure, particularly in the health and medical sectors, would reduce the deaths caused by pandemics like COVID-19.

(5) It is critical to reduce illiteracy and raise public awareness as it is associated with the number of deaths caused by epidemics like COVID-19. This would be critical for Tehran’s southern half, where illiteracy is prevalent.

### 4.2. Limitations and Futures Research Strategy

This study has a few limitations. The COVID-19 mortality data were obtained from the Ministry of Health’s Health Information System (HIS). COVID-19 disease was initially difficult to recognize, and its specifics were not documented in hospitals. As a result, some of the first patients may have yet to be fully documented. Furthermore, due to a lack of health care during the peak of the COVID-19 epidemic, many patients may have died outside of hospitals and were not recorded in the Ministry of Health’s HIS system. Another limitation is the need for more local household income data. We attempted to avoid this issue by examining other socioeconomic indicators, such as the unemployment rate. Additionally, due to sanctions and economic difficulties, most of Tehran’s power plants have used low-quality fuel oil for decades [121]. The use of this oil might release other air pollutants that we have not considered in this study since monitoring stations have never measured other potential air pollutants in Tehran. Instead, we looked at other significant air pollution indicators. Additionally, it is worth mentioning that we had not controlled for the number of days to calculate the monthly mortality rates to create Figure 4.

Furthermore, the deceased’s geographic coordinates were not recorded. As a result, aggregated statistics were used at the neighborhood level. Future research could employ GIS techniques to predict COVID-19 mortality rates in Tehran and develop a mortality risk map based on neighborhood characteristics. In addition, more research is needed to determine how COVID-19 mortality rates in metropolitan areas are associated with determinants, such as underlying conditions, employment type, income level, place of work address, and other information about the deceased’s health status.

## 5. Conclusions

A spatiotemporal pattern of the COVID-19 mortality rate in Tehran was investigated, with neighborhoods having higher mortality rates at different times. COVID-19 mortality rates were found to be significantly related to the Air Quality Index, age, and illiteracy rates, and these associations varied across Tehran neighborhoods. The MGWR model showed the best performance among the different regression models used to identify factors associated with COVID-19 mortality rates. Our approach could be useful for future COVID-19 mortality rate modeling and other infectious diseases. Finally, the findings have implications for COVID-19 mortality rate reduction initiatives in long-term urban health plans that promote healthy and resilient cities.

## Figures and Tables

**Figure 1 tropicalmed-08-00085-f001:**
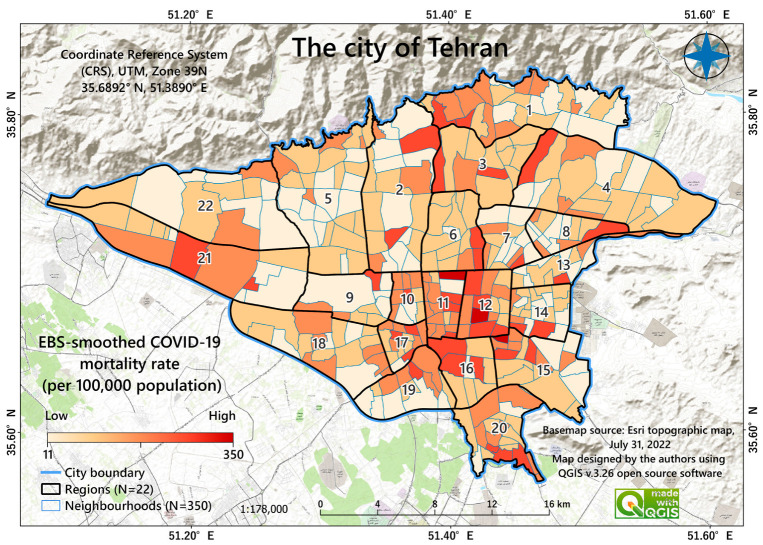
Map of Tehran, Iran, showing the spatial distribution of COVID-19 mortality 2019–2021; EBS = empirical Bayes smoothed mortality rates per 100,000 population at the neighbourhood level. Numbers indicate the administrative district division of the city. Each district includes some neighbourhoods.

**Figure 2 tropicalmed-08-00085-f002:**
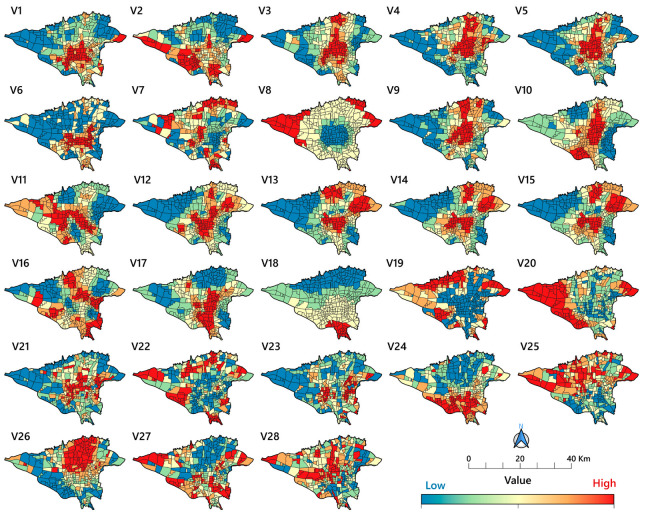
Spatial distributions for selected explanatory variables in Tehran, Iran. Dark blue shades show low ranges and dark red shades show high range values for each variable (numbers as given in Table 1).

**Figure 3 tropicalmed-08-00085-f003:**
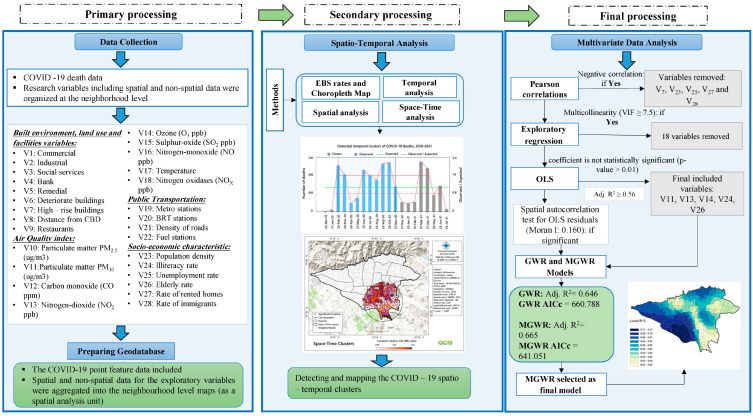
Methodology flowchart of the study of COVID-19 mortality in Tehran.

**Figure 4 tropicalmed-08-00085-f004:**
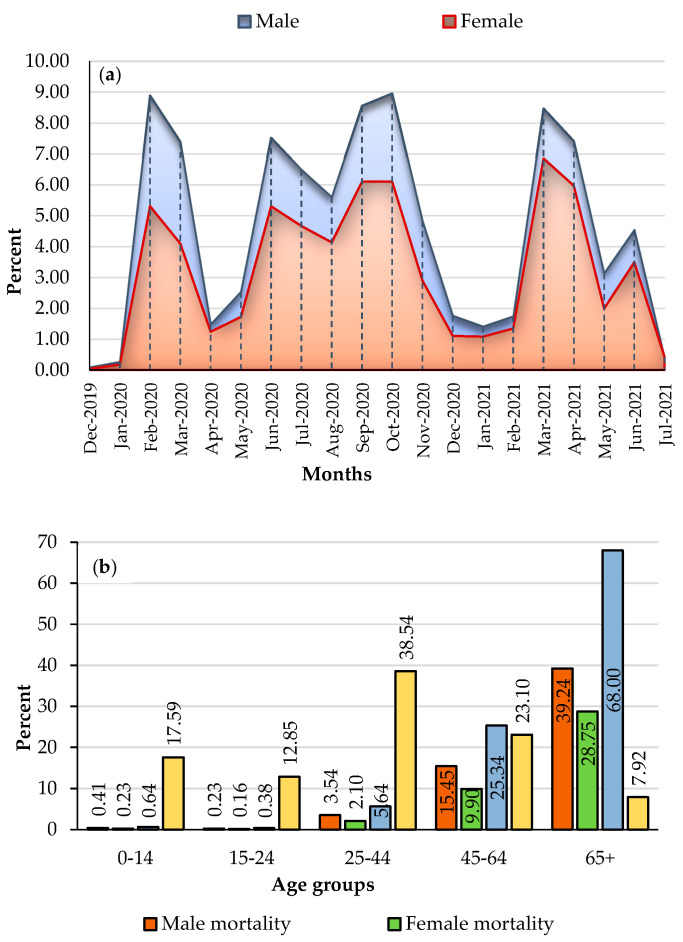
(**a**): Monthly distribution chart of the mortality rates (per 100,000 population) by number and sex; (**b**): percentage of COVID-19 related deaths by number, sex, and age group.

**Figure 5 tropicalmed-08-00085-f005:**
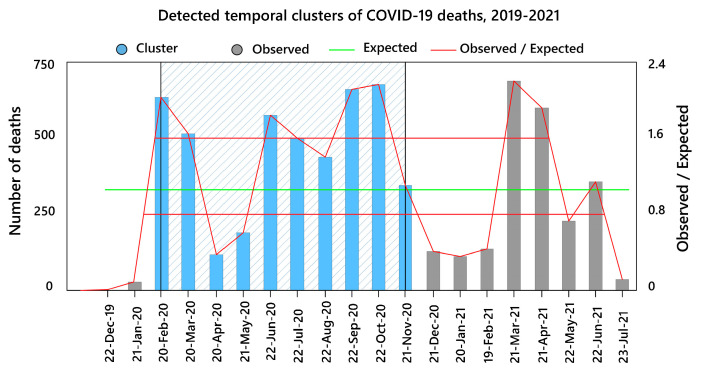
Purely temporal significant clusters of COVID-19 deaths during 2019 to 2021 in the study area.

**Figure 6 tropicalmed-08-00085-f006:**
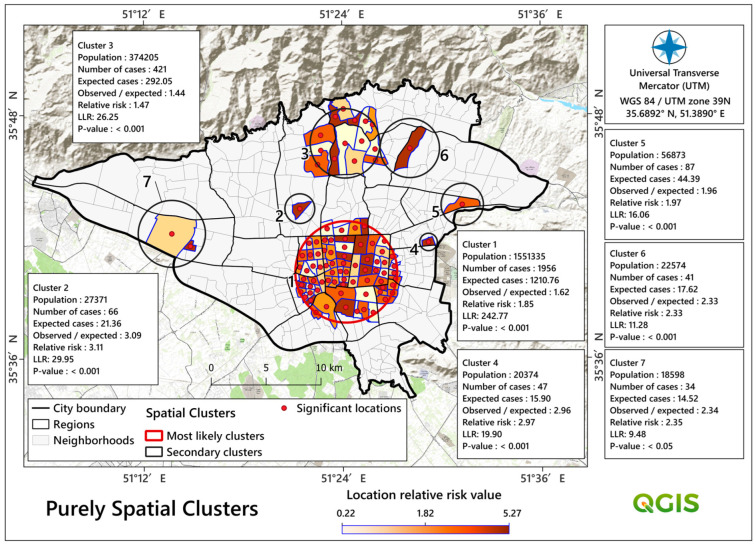
Detected purely spatial clusters of COVID-19 deaths in Tehran, Iran.

**Figure 7 tropicalmed-08-00085-f007:**
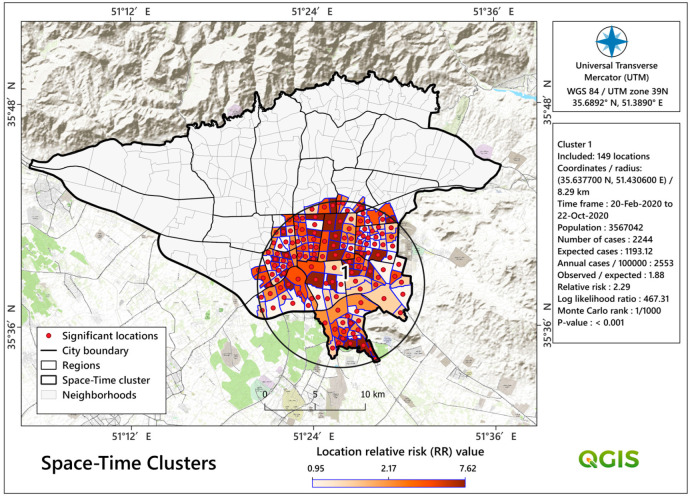
Detected space-time clusters of COVID-19 deaths in Tehran, Iran.

**Figure 8 tropicalmed-08-00085-f008:**
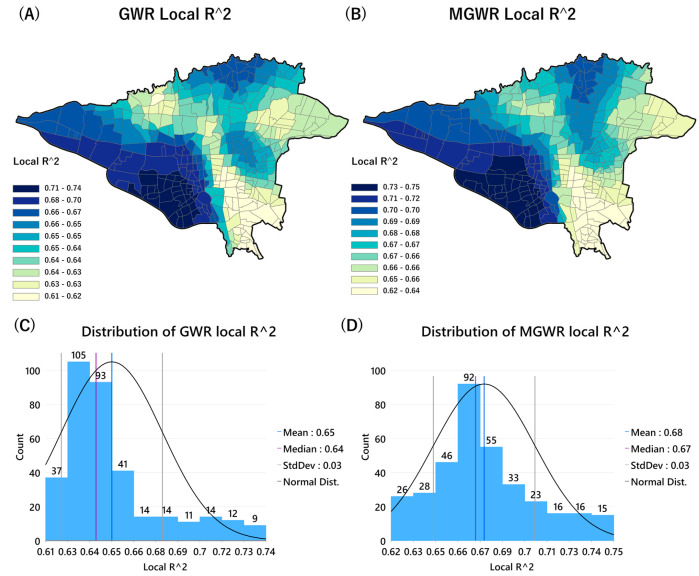
Spatial distribution of local R^2^ values for GWR and MGWR models. (**A**): Map of the GWR local R^2^ values, (**B**): Map of the MGWR local R^2^ values, (**C**): Histogram chart depicting the distribution of GWR R^2^ values; and (**D**): Histogram chart depicting the distribution of MGWR R^2^ values in the study area.

**Figure 9 tropicalmed-08-00085-f009:**
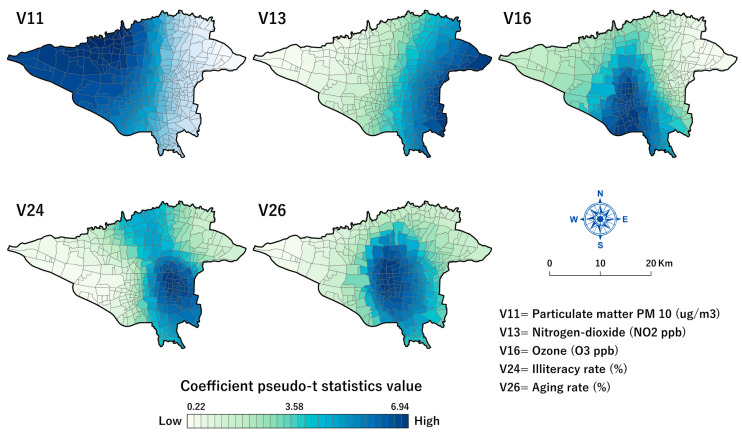
Surface map of local estimates pseudo t-values of MGWR model results for Tehran’s COVID-19 mortality rate dataset. Dark blue shades show areas with high t-values. All maps were generated in ArcGIS Pro 3.0.2 (ESRI, Redlands, CA, USA, 2022).

**Figure 10 tropicalmed-08-00085-f010:**
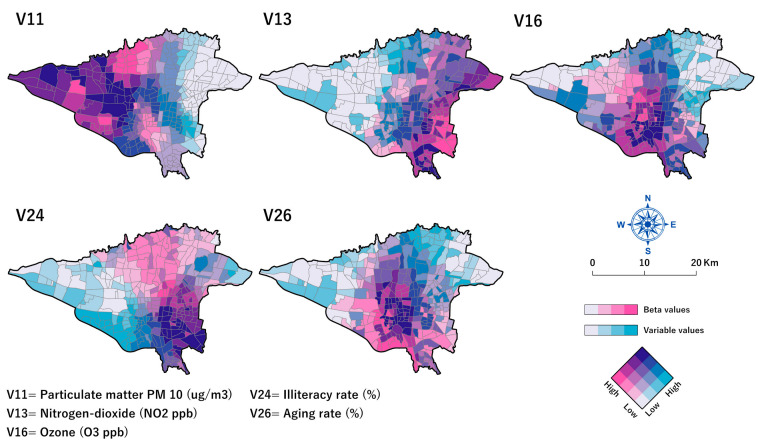
Spatial distribution bivariate map of local estimates (*Beta*) and covariates original values of MGWR model results for Tehran’s COVID-19 mortality rates. All the maps were generated in ArcGIS Pro 3.0.2 (ESRI, Redlands, CA, USA, 2022).

**Table 2 tropicalmed-08-00085-t002:** Purely spatial detected clusters based on a Poisson scan statistic model for areas with high rates of COVID-19 deaths in Tehran, Iran.

Cluster	Neighbourhood	Coordinates/Radius	Population	Cases (No.)	Expected Cases (No.)	Annual Cases/100,000 (No.)	Observed/Expected Cases (No.)	Relative Risk	Log Likelihood Ratio	Monte Carlo Rank	*p*-Value
1	Includes seventy-one neighbourhoods	(35.671100° N, 51.403600° E)/4.63 km	1,551,335	1956	1210.76	2192.9	1.62	1.85	242.78	1/1000	<0.001
2	Includes one neighbourhood	(35.723200° N, 51.357100° E)/1.3 km	27,371	66	21.36	4193.9	3.09	3.11	29.95	1/1000	<0.001
3	Includes fourteen neighbourhoods	(35.776700° N, 51.402900° E)/3.17 km	374,205	421	292.05	1956.7	1.44	1.47	26.25	1/1000	<0.001
4	Includes one neighbourhood	(35.695400° N, 51.486500° E)/0.8 km	20,374	47	15.90	4012.2	2.96	2.97	19.9	1/1000	<0.001
5	Includes one neighbourhood	(35.726200° N, 51.520600° E)/1.87 km	56,873	87	44.39	2660.6	1.96	1.97	16.06	1/1000	<0.001
6	Includes one neighbourhood	(35.772600° N, 51.468100° E)/2.6 km	22,574	41	17.62	3158.9	2.33	2.33	11.28	3/1000	<0.001
7	Includes two neighbourhoods	(35.702800° N, 51.228200° E)/2.05 km	18,598	34	14.52	3179.6	2.34	2.35	9.48	25/1000	<0.001

A cluster is statistically significant when its log likelihood ratio is greater than the critical value, which is, for the significance level: Gumbel Critical Values: 0.00001: 18.16;… 0.0001: 15.57; … 0.01: 5.21. Standard Monte Carlo Critical Values: 0.001: 12.545047; 0.01: 10.568609; 0.05: 8.628534.

**Table 3 tropicalmed-08-00085-t003:** Model specifications and diagnostic metrics for the fitted GWR model.

Diagnostic Name	Value		Value
Residual sum of squares	114.911	AICc	660.788
Effective number of parameters (trace(S))	25.428	BIC	758.252
Degree of freedom (n–trace(S))	324.572	R^2^	0.672
Sigma estimate	0.595	Adj. R^2^	0.646
Log-likelihood	−301.718	Adj. alpha (95%)	0.012
Degree of dependency (DoD)	0.753	Adj. critical t value (95%)	2.531
AIC	656.293		-

**Table 4 tropicalmed-08-00085-t004:** Summary statistics of the GWR and the MGWR coefficients.

**GWR Model**						
*Variable*	*Bandwidth*	*Mean*	*STD*	*Minimum*	*Median*	*Maximum*
Intercept	172	−0.108	0.176	−0.399	−0.125	0.217
PM_10_	172	0.118	0.046	0.001	0.124	0.222
NO_2_	172	0.332	0.164	0.055	0.340	0.616
O_3_	172	0.319	0.123	−0.136	0.318	0.559
Illiteracy rate (%)	172	0.139	0.129	−0.074	0.126	0.377
Ageing rate (%)	172	0.369	0.084	0.195	0.368	0.654
**MGWR Model**						
*Variable*	*Bandwidth*	*Mean*	*STD*	*Minimum*	*Median*	*Maximum*
Intercept	59	−0.049	0.280	−0.545	−0.070	0.620
PM_10_	348	0.115	0.008	0.103	0.113	0.129
NO_2_	335	0.375	0.030	0.311	0.384	0.408
O_3_	253	0.327	0.053	0.236	0.317	0.422
Illiteracy rate (%)	196	0.125	0.065	0.013	0.130	0.241
Ageing rate (%)	348	0.374	0.004	0.365	0.374	0.383

**Table 5 tropicalmed-08-00085-t005:** Model specifications and diagnostics metrics for the fitted MGWR model.

Diagnostic Name	Value		Value
Residual sum of squares (RSS)	108.666	AICc	641.051
Effective number of parameters (trace (S))	25.352	BIC	738.247
Degree of freedom (n–trace (S))	324.648	R^2^	0.690
Sigma estimate	0.579	Adj. R^2^	0.665
Log-likelihood	−291.940		
Degree of dependency (DoD)	0.754		
AIC	636.583		-

**Table 6 tropicalmed-08-00085-t006:** Model comparison and performance diagnostics.

Model	RRS	Log-Likelihood	AIC	AICc	R^2^	Adj. R^2^
OLS	151.110	−349.642	1027.85	3455.81	0.568	0.561
GWR	114.911	−301.718	656.293	660.788	0.672	0.64
MGWR	108.666	−291.940	636.583	641.051	0.690	0.665

## Data Availability

All the data used in this study are available via Appendix A.

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
