# Peer review of "The COVID-19 Mortality Rate Is Associated with Illiteracy, Age, and Air Pollution in Urban Neighborhoods: A Spatiotemporal Cross-Sectional Analysis"

_tropicalmed, 2023, doi:10.3390/tropicalmed8020085_

Round 1

Reviewer 1 Report

please see attached file!

Author Response

Dear Reviewer,

We would like to thank you for spending a lot of time on our manuscript. We did our best to address most of the comments in this round of revision. ‎Here, we have provided a point-by-point response to your comments and we have mentioned any corresponding changes in the manuscript.

We have ‎provided both track-changed and clear versions for our revision; however, the line and ‎page numbers in this response letter have been written according to the clear version ‎of the revision.‎  

Reviewer #1:

Comment 1: Still, for a simple-minded person like me, it is difficult to understand the findings of the geographically weighted regression models. I would want to learn how a certain environmental or socioeconomic factor is impacting on risk of COVID-19 death, not just in the East or in the West of Tehran, but in general. How am I to interpret a coefficient that has a certain value in one part of the city and quite a different one in another? Maybe some guidance for the simple reader could be helpful?

Authors' Response: Thanks for this comment. The general coefficients for both GWR and MGWR models have been provided in table 4 (column “mean”) in the manuscript. For example, the general coefficient of NO2 in the MGWR model is 0.375. According to this comment, we have added the following explanation to the manuscript to clarify this potential question for the readers (section 3.3.2 GWR model results).

The column “Mean” in table 4 shows the general coefficients of both GWR and MGWR models. For instance, NO2 had the strongest association with COVID-19 mortality rates in the whole city, regardless of any specific geographical area.”

Comment 2: I seriously do appreciate the many figures and all the information presented by the authors. But are really all results necessary? I do understand the statistical issue about clustering. If I’d analyze spatial and temporal variation of any non-communicable disease, I would be interested in clustering. And with each cluster, I would ask myself: “what might be the reason?” But infectious diseases are – well – infectious. That means they tend to cluster: a single case tends to breed new cases. This is the core characteristic of “infectious”. I’d be rather surprised if infectious diseases would not cluster. And if I see a cluster, I would not wonder if I could find any deeper environmental causes for that cluster, but assume that clustering is just the normal thing for infectious diseases. COVID-19 did occur in waves. The authors count 4 waves in total in Tehran. But only the first three waves together form a single temporal cluster according to their statistics. I find that “wave” in the case of an epidemic or pandemic has much more biological relevance than the statistical cluster. In studies of infectious diseases, the term “cluster” is even used in a completely different sense: This denotes cases linked together by individual linkage: Each case of a cluster must have been proven to have met another case at the right time. Eventually, even sequencing is used to prove that the cases are connected biologically, not just statistically. So, what is the gain from reporting temporal clusters instead of focusing on the waves?

Authors' Response: Thanks for this comment. We do agree that in the context of infectious diseases, having clusters is not surprising, and they are usually formed due to the nature of infectious diseases such as COVID-19. However, two essential points are worth mentioning here. First, these clusters are identified according to COVID-19 mortality rates, not the prevalence rates which are the basis for the presented waves. As you said, our analysis identified that the first three waves are a mortality cluster, but the fourth wave is not. We believe that the effect of vaccines and other preventive strategies result in not being included the wave four in our time cluster. Second, according to our data, we aimed to show the real mortality waves. We agree that clustering for infectious diseases is necessary, but we aimed to know when these clusters were formed.

Comment 3: In the discussion (line 537 and following), the authors state: “As a result of this variability and clustering, COVID-19 mortality clusters must have been influenced by the particular characteristics of the various metropolitan districts that was our second aim in this study.” Allow me to disagree! As I said before: if we would talk about non-communicable diseases, for example mesothelioma, cancer of the nasal cavity, skin cancer of the scrotum, hemangio-sarcoma of the liver, to name but a few classical examples, any cluster would make me seek environmental or occupational causes. But a cluster of infectious diseases? No! Infectious diseases will be clustered because they are infectious. Therefore, I beg the authors to reconsider at least their take on temporal clustering.

Authors' Response: Thanks for this comment. We agree with the reviewer that the word “must” in this sentence is too strong and replaced it with “might”. However, please note that this sentence not only talks about temporal clusters but also expresses the ideas regarding spatial clusters we found in the city.

Comment 4: Line 41: when you provide the total number of cases in Iran (“7, 538, 125”): (a) if you want to provide an exact number, please do not leave a space after each comma. It is unusual and the figure is difficult to read. At first, I thought you talked about 3 different figures. (b) I doubt an exact number can ever be provided. Maybe “more than 7 million” would be much more accurate and informative enough. But this is entirely up to you!

Authors' Response: Done. We have written “more than 7 million” according to your suggestion. Also, in other cases, we removed any additional spaces between the commas.

Comment 5: Line 81/82: “Spatial regression methods such as GWR were used to address non-stationary.” Either a noun is missing (“non-stationary distribution of parameters”, for example), or you should write “non-stationarity” instead, if this is what you wanted to say!

Authors' Response: Done. The word revised (Line 82).

Comment 6: Line 114-118: I struggle with the numbers: 21,503 persons (per km²) times 730 km² is far more than about 9 million people. (20,000 x 700 would already be 14 million!) Either the average is incorrect (maybe you failed to do weighted averaging?) or some of the 730 km² are not part of the settled area. Could you please check and explain?

Authors' Response: Many thanks for your careful consideration. We had brought these numbers from other sources, but we had access to the original data and recalculated and modified it (Lines 115-118).

Comment 7: Line 154: “The socioeconomic characteristics of neighborhoods (including unemployment rates) were provided by the Iranian Statistical Centre in 2016.” I might be mistaken. But I guess the better way would be to say: “The socioeconomic characteristics of neighborhoods (including unemployment rates) of 2016 were provided by the Iranian Statistical Centre.” (If I understand correctly, these were 2016 data published in 2022)

Authors' Response: Thank you for this comment. Done. (Lines 155-156)

“In addition, the Iranian Statistical Centre provided the socioeconomic characteristics of neighborhoods (including unemployment rates) of 2016”.

Comment 8: Line 183: “between six such variables (population density, illiteracy, unemployment, older age, rented home, immigrant, and neighbourhood-level COVID-19 mortality).” The closing bracket is misplaced! Write: “between six such variables (population density, illiteracy, unemployment, older age, rented home, and immigrant) and neighbourhood-level COVID-19 mortality.”

Authors' Response: Thank you for this comment. Done. (Line 187)

Comment 9: Table 1. Sorry, I am an expert in environmental health, very much interested in air pollution. So, I wonder: You write the pollutants’ concentrations were provided per km². Therefore, I do believe that the data come from a dispersion model or something similar that uses a 1 km x 1 km grid. But in the end, you use concentration per neighborhood, not per km². And one neighborhood could partly overlap several squares of that grid. How did your average squares across neighborhoods? And I also wonder, especially regarding ozone: Regarding health effects, short term peaks of ozone are often much more relevant than daily means. So, epidemiological research, instead of using daily means, usually uses daily maximal (8-hour or 1-hour) means. And even in the case of chronic exposure, the studies usually use the average of these maximums. And lastly, concentrations vary over time. I suppose you used annual averages, yes? From which year?  

Authors' Response: Thank you for this invaluable comment. First, we extracted available data of the annual average of air pollutants for five recent years, which Air Pollution monitoring stations in the study area gathered. Then we mapped the pollution using interpolation methods in GIS. That method gave us precise but estimated values in each cell (1 km x 1 km grid). The result was a raster map. Then because our scale of analysis was neighbourhoods, we calculated the average pollution values for each neighbourhood by using zonal statistics and extraction tools in GIS. For your comment, we revised the text and used a 1 km x 1 km grid instead of a square km. We made minor changes in table 1 and added a paragraph to the method section as follows. (Lines 173-178)

 “In this study, the average of 5 years (2016 to 2021) for each pollutant was derived from the pollutant-related data of air pollution monitoring stations. Then, using the inverse distance weighted (IDW) interpolation technique, data were calculated in a GIS with pixels 1 x 1 km in size. The 5-year average of pollution was then calculated for each neighbourhood separately using zonal statistical methods.”

Comment 10: Line 207: “The log-likelihood ratio (LLR) and relative risk (RR) were calculated to compute a p-value...” I believe you should write: “The log-likelihood ratio (LLR) and relative risk (RR) were calculated. To compute a p-value (Monte Carlo simulations … were used).”

Authors' Response: Thank you for this comment. The sentence has been revised as follows. (Lines 212-215)

“Monte Carlo simulations, first introduced by Dwass, [83] were utilized to calculate a p-value”

Comment 11: Line 218: “The Monte Carlo testing, which was set to calculate test statistics for each random replication at the p=0.05 level.” This is not a complete sentence. Maybe you want to say: “The Monte Carlo testing was set to calculate test statistics for each random replication at the p=0.05 level.”

Authors' Response: Thank you for this comment. The sentence has been revised as follows:

“The Monte Carlo testing was set to calculate test statistics for each random replication at the p=0.05 level.”

Comment 12: Line 281 and thereafter: what is the meaning of superscript e.g. in “(ui, vi)” or “ß0(ui, vi)”?

Authors' Response: Thank you for this comment. The issue was raised when we used MathType to write the formulas. It has been corrected now.

Comment 13: Figure 4 a: I am not sure about the x-axis. It is numbered “1-12 (curious symbol)-19” and so on. I assume that this stands for “December 1, 2019” (and the symbol somehow stands for “month”). If this is correct, then December 2019 is between the first tick (Dec 1) and the second tick (January 1). But you show monthly data. So how could the rate increase between the two dates? Or the first tick just means “December 2019”. But what then is the meaning of the number “1” at the beginning? And, in the text, (line 323), the months are simply numbered, not named (“months (e.g., 3, 10, 11, and 16)”). Month 3 would then be February 2020. It is difficult to switch from the text to the figure and back. But anyway: these are rates per month. That means absolute numbers per month divided by population. But different months have different length. Even if more people died in February than in January per day, there could still be more deaths in January than in February. Did you control for that? Better calculate and display average daily rate per month!

Authors' Response: Thank you again for careful consideration. We transferred data and charts from Microsoft Excel to Microsoft Word. Then surprisingly in Word, the numbers were changed to incorrect format. For example, 22 December was changed to 01 December and so on. In the new version, Figure 4 revised and we have corrected all the issues. According to your comment, we have also corrected the text. (Line 333). Finally, we have to say that we had not controlled for the number of days to create figure 4, so we have added it to the limitation section of our manuscript as follows:

“Also, it is worth mentioning that we had not controlled for the number of days to calculate the monthly mortality rates to create figure 4.”

Comment 14: Figure 4b: shows the distribution of all deaths among age groups and sexes. But: (a) age groups span different numbers of years. The first age-group spans 15 years, the second 10. Then two age groups of 20 years each, and last, there is an age group with open end. Maybe there are just many more elderly people and therefore you have more deaths in this age group? The higher mortality and lethality with increasing age is plausible, but it is not correctly displayed with this figure! And sorry, I have completely no idea what that dotted line (“2 per. Mov. Avg. (Total)”) signifies!

Authors' Response: Thanks for this comment. We have added a new column to this figure showing the total population percentage in each age group to address your comment. As you can see, the older population percentage is less than other age groups but there are more deaths in this group. Also, the dotted line has been removed according to your comment.

Comment 15: Line 393: “our model accounts for about of the COVID-19 mortality rates”: Here I miss the percentage: “about 66%”. But then even a simple mind like mine understands the meaning of R²!

Authors' Response: Done. Text has been revised (Line 404)

Comment 16: Line 540: “Previous research indicates that environmental crises, such as air pollution concentration in Tehran...” Do you really mean “crises”? Wouldn’t “pollution”, “exposures”, or “problems” be a better word?

Authors' Response: Thank you for all helpful comments. We have replaced the “crisis” with the “problems”.

Comment 17: Line 572: “In accordance with earlier studies revealed the association between ozone and high COVID-19 mortality rates...” Please write: “earlier studies that revealed the association”!

Authors' Response: Done. Text have revised. (Line 585)

Comment 18: Line 544 and following is just one example. The same discussion is then also provided for the other pollutants: “PM10 was substantially associated with COVID-19 mortality rates in the majority of neighbourhoods on the western side of the city where the PM10 levels were similarly elevated. This can be explained by the fact that many factories (e.g., Industrial Sand and Cement Factories) are located in this area of the city.” Factories explain the high PM levels in the west. But do they also explain the substantial association? Why is this the case? What do you want to say? I see two options: (a) “PM from factories is more detrimental than PM from other sources. Therefore, the PM is more substantially linked to COVID-19 mortality in the region where a higher amount of PM comes from industries.” or (b): “Factories are in the west. Therefore, in the west are higher PM levels. Higher PM levels are closer or more strongly associated with COVID-19 mortality. This shows that the association between PM and mortality is not linear and maybe indicates a no-effect threshold.” Both statements would be interesting, but would call for more elaboration. Just stating that association with mortality AND high levels are explained by industry, is not sufficient. By the way: Against statement (b) stands the observation of weak association in spite of high PM levels in the north! As I explained in the beginning: My simple mind calls for help with the interpretation of coefficients that vary across regions. And these sentences in the discussion simply are not helpful. They pick out the regions where both association and concentrations are high but neglect the regions where either associations or concentrations are high, but not both.

Authors' Response: Thanks for this detailed comment. First, as you mentioned correctly, this kind of analysis shows us four different groups in terms of the strength of the association and the amount of the air pollutant gas: High association with the high level of pollutant, High association with the low level of pollutant, low association with the high level of pollutant, and low association with the low level of pollutant. Although a high amount of an air pollutant is not good at all and is always important, in our specific study, highly correlated areas are important to consider as they have a big effect on mortality even with a low level of a certain air pollutant. As a result, it is important to discuss two groups: high association with a high level of an air pollutant and high association with a low level of the pollutant. If we wanted to talk about it in simple words, high association areas are important to discuss regardless of the value of variables. We have discussed these areas in the discussion part. Regarding the general effect of the explanatory variables on COVID-19 mortality, as we said before, it has been mentioned in Table 4 (column “mean”).

According to this comment, we have revised the discussion part in different places.

Comment 19: Line 608: “Many old people cannot take care of their health properly and if they are also subject to the harmful air of the air quality in central Tehran.” Why “if”? If you write “if”, then I expect a “then”.

Authors' Response: Done. The text has been revised.

Comment 20: Line 616: What do you mean by “climatic pollutants”? Air pollutants? Atmospheric pollutants? Can you pollute the climate?

Authors' Response: Done. The text has been revised and “air pollutants” has been replaced with “climatic pollutants”.

Comment 21: Line 647: “most of Tehran's power plants have used low-quality fuel oil for decades due to sanctions and economic difficulties [121] that caused pollution has never been assessed.” The sentence sounds grammatically wrong. But what do you mean by “has never been assessed”? Do you want to say that these power plants emit pollutants that are not covered by PM, NO2 etc.? Which pollutants would that be? Mercury? And what do you mean by “assessed”? They have not been measured (where? At the stack, at monitoring stations?) / The health-effects have not been studied (in Tehran? Anywhere in the world?) Please reconsider the whole sentence!

Authors' Response: Thanks for this comment. We did our best to clarify this part in the light of your comments. The text has been revised as follows:

“Additionally, due to sanctions and economic difficulties, most of Tehran's power plants have used low-quality fuel oil for decades [121]. The use of this oil might release other air pollutants we have not considered in this study since monitoring stations have never measured other potential air pollutants in Tehran. Instead, we looked at other significant air pollution indicators.”

We wanted to say this as we think there are other gases besides the ones we examined in this study.

Best regards,

Authors,

Reviewer 2 Report

The paper: The COVID-19 mortality rate is associated with the levels of illiter3 acy, age, and air pollution in urban neighborhoods.: A spatial and temporal cross-sectional analysis represent very detailed and serious study, which is of the interest of scientific and research community. 

This study aimed to look into the spatial and temporal trends of COVID-19 mortality in Tehran, Iran, and their associations with socioeconomic, air quality, public transportation, and built environment variables,

Methodology is very well described. 

Results are very detailed, clearly presented. 

The authors gave Policy implications, as well as limitations of the study. 

The main policy implication is that itis critical to reduce illiteracy and raise public awareness as it is associated with

635 the number of deaths caused by epidemics like COVID-19. This would be critical for Teh636 ran's southern half, where illiteracy is prevalent.

They underline, among other, that ore research is needed to determine

 how COVID-19 mortality rates in metropolitan areas associated with determinants such as underlying conditions, employment type, income level, place of work address, and other information about the deceased's health status.

Ethical approval is given - The study was conducted in accordance with the Mohaghegh Ardabili University and approved by the Research Ethics Committee of Shahid Beheshti University, Tehran, Iran (#IR.SBU. RE683 TECH.REC.1399.076) for studies involving humans.

Suggestion - to move aim of the study at the end of introduction from discussion. 

English check and grammar is suggestion, too. 

Author Response

Dear Reviewer

We would like to thank you for spending a lot of time on our manuscript. We did our best to address most of the comments in this round of revision. ‎Here, we have provided a point-by-point response to your comments and we have mentioned any corresponding changes in the manuscript. We have ‎provided both track-changed and clear versions for our revision; however, the line and ‎page numbers in this response letter have been written according to the clear version ‎of the revision.‎  

Reviewer #2:

Comment 1: Suggestion - to move aim of the study at the end of introduction from discussion.  

Authors' Response: Thanks for this comment. Done.

Comment 2: English check and grammar are suggestion, too.

Authors' Response: Thanks for your suggestion. We did an English proofreading in the revision step.

Best regards, 

Authors,

Round 2

Reviewer 1 Report

I agree with the responses of the authors and thank them for their considerations. The authors have made also other changes to the text that mostly are fine. (I use the version with track changes to provide line numbers.)

Line 143:

Original text: “We excluded data that were incomplete or inaccurate (about 50 records).“

New text: “We excluded the data that were needed to be completed or accurate (about 50 records).“

I believe the original version was better. Why did you exclude accurate data? “needed to be completed“ says the same as “incomplete“, but is a much more complicated text.

Author Response

Dear Reviewer,

Thank you very much for the review of our manuscript. We sincerely appreciate all the valuable comments and suggestions, which helped us to improve the quality of the article.

We revised the phrase as below and highlighted it by the red color in the main text of our revised manuscript: 

Page 4, lines: 137-138.

Best regards,

Corresponding author,

On behalf of all the authors,

Reviewer 2 Report

All comments and suggestion have considered and paper is improved. 

Author Response

Dear Reviewer,

Thank you very much for the review of our manuscript. We sincerely appreciate all the valuable comments and suggestions, which helped us to improve the quality of the article.

Best regards,

On behalf of all the authors,